

# Improving Cloud Type Classification of Ground-Based Images Using Region Covariance Descriptors

Yuzhu Tang[1, 3], Pinglv Yang[2], Zeming Zhou[2], Jianyu Chen[3], Delu Pan[3], Xiaofeng Zhao[2]

[1] School of Computer Science and Engineering, Nanjing University of Science and Technology, Nanjing, 210094, China
[2] College of Meteorology and Oceanology, National University of Defence Technology, Nanjing, 211101, China
[3] State Key Laboratory of Satellite Ocean Environment Dynamics, Second Institute of Oceanography, Hangzhou, 310012, China

*Correspondence to*: Zeming Zhou (zhou_zeming@nudt.edu.cn), Jianyu Chen (chenjianyu@sio.org.cn)

**Abstract.** Cloud types are important indicators of cloud characteristics and short-term weather forecasting. The meteorological
researchers can benefit from the automatic cloud type recognition of massive images captured by the ground-based imagers.
However, by far it is still of huge challenge to design a powerful discriminative classifier for cloud categorization. To tackle
this difficulty, in this paper, we present an improved method with region covariance descriptors (RCovDs) and Riemannian
Bag-of-Feature (BoF). RCovDs model the correlations among different dimensional features, that allows for a more
discriminative representation. BoF is extended from Euclidean space to Riemannian manifold by $k$-Means clustering, in which
Stein divergence is adopted as a similarity metric. The histogram feature is extracted by encoding RCovDs of the cloud image
blocks with BoF-based codebook. The multi-class support vector machine (SVM) is utilized for the recognition of cloud types.
The experiments on the ground-based cloud image datasets validate the proposed method and exhibit the competitive
performance against state-of-the-art methods.

## 1 Introduction

Clouds have a strong impact on the Earth's energy budget balance, climate modeling and weather prediction. Cloud type
variations (e.g., variations in cloud-top height and water content) may affect both shortwave and longwave radiative fluxes.
During climate variations, the distribution and frequency of different cloud types may change (Chen et al., 2000). Additionally,
accurate cloud classification, especially convective cloud identification, are essential to Hazardous weather monitoring (Zhang
et al., 2018a). In recent years, the growing appeal on renewable solar energy pushes additional interest on cloud coverage
measurement and cloud classification (Heinemann et al., 2006; J. Huertas, 2017; Martínez-Chico et al., 2011). Therefore,
accurate cloud type classification is in great need. Currently, the classification task is mainly undertaken by manual observation,
which is labor-intensive and time-consuming. Benefiting from the development of ground-based cloud image devices, we are
able to continuously acquire cloud images and automatically classify the cloud types.

Clouds are by their very nature highly variable (Joubert, 1978), which makes the automatic classification a tough task. It is
found that structure and texture are suitable to describe the visual appearance of clouds. The structural features include intensity



gradient (Luo et al., 2018), mean grey value (Calbó and Sabburg, 2008; Liu et al., 2011), the census transform histogram (Xiao et al., 2016; Zhuo et al., 2014), edge sharpness (Liu et al., 2011), and features based on Fourier transform (Calbó and Sabburg, 2008). The textural features contain the scale invariant feature transform (SIFT) (Xiao et al., 2016), the grey level co-occurrence matrix (GLCM) (Cheng and Yu, 2015; Heinle et al., 2010; J. Huertas, 2017; Kazantzidis et al., 2012; Luo et al., 2018), the local binary patterns (LBP) (Cheng and Yu, 2015) and its extensions (Liu et al., 2015; Wang et al., 2018b). Commonly, no single feature is best suited for cloud type recognition, thus most existing algorithms tend to integrate multiple features to describe the cloud characteristics. However, those algorithms rarely consider the correlations between different dimensional features, which could lower the classification accuracy.

Within recent years, convolutional neural networks (CNNs) have been exploited to tons of image recognition and has achieved remarkable performance (Krizhevsky et al., 2012). Being different from hand-crafted features, CNNs extract hierarchical features including the low-level details and high-level semantic information. Recently, plenty of works (Shi et al., 2017; Ye et al., 2017) have obtained encouraging results by extracting the cloud signature from pre-trained CNNs, such as AlexNet (Krizhevsky et al., 2012) and VGGNet (Simonyan and Zisserman, 2015). In addition, attempts have been made to simply exploit end-to-end CNN models for cloud categorization (Li et al., 2020; Liu et al., 2019; Liu and Li, 2018; Liu et al., 2018; Zhang et al., 2018b). However, the insufficiency of labelled samples might make the network hard to converge in the training stage.

The main challenges of the ground-based cloud image classification task can be ascribed to the following reasons: (1) One single feature cannot effectively describe different types of clouds, we need to extract textural, structural, and statistical features simultaneously. (2) The scale of cloud varies greatly, therefore, the extracted features should be robust in the presence of illumination changes and nonrigid motion. (3) Different cloud types may have similar local characteristics, and thus the global features need to be considered. To address those issues, we utilize the region covariance descriptors (RCovDs) to encode the features of the cloud image blocks, and with the aid of Bag-of-Feature (BoF), we aggregate those local descriptors to obtain the global cloud image feature for cloud type classification.

The performance of RCovDs (Tuzel et al., 2006) is proved to be superior on object detection (Carreira et al., 2015; Guo et al., 2010; Li et al., 2013; Pang et al., 2008) and classification tasks (Fang et al., 2018; Li et al., 2013; Wang et al., 2012). As the second-order statistics of the image features, RCovDs can provide rich and compact context representations. The noises are largely filtered out by removing the mean values of the features. RCovDs are also scale and rotation invariant, irrespective of the pixel positions and numbers of sample points. Despite of their attractive properties, directly adopting RCovDs for cloud type classification is still of difficulty on account of their non-Euclidean geometry property. RCovDs are Symmetric Positive Defined (SPD) matrices and naturally reside in a Riemannian manifold, therefore, the machine learning algorithms on Euclidean space should be adapted for the automatic cloud image recognition.

In Euclidean space, BoF describes an image as a vector from a set of local descriptors (Jégou et al., 2012), and it aggregates the local features to obtain a global representation. Inspired by the work in (Faraki et al., 2015a), we encode RCovDs of the





local image blocks into a histogram by using Riemannian counterpart of the conventional BoF, taking the geodesic distance

of the underlying manifold as the metric.

In this paper, we extend our previous work (Luo et al., 2018), and propose an improved cloud type classification method based on RCovDs. The diagram is shown in Fig. 1. In the first step, we extract multiple pixel-level features such as intensity, color and gradients from the cloud image blocks to form RCovDs. In the second step, RCovDs are encoded by the Riemannian BoF to output the histogram representation. In the last step, the histogram is taken as the feed of the multiclass SVM for cloud

type prediction.

The main contributions of this paper are:

- The RCovD is firstly introduced to characterize the cloud image local patterns and the Riemannian BoF is applied to encode RCovDs into image-level histogram;

- The impacts of Riemannian BoF codebook size and the image block size on cloud type classification accuracy are

investigated;

- For The small training dataset, the proposed algorithm offers better performance as compared to the state-of-the-art approaches.

The remainder of this paper is organized as follows. Section 2 introduces the ground-based cloud image datasets and details the proposed cloud type classification method. Experimental results and comparisons with other methods are presented in

Section 3. Section 4 concludes our contributions and discusses the future work.

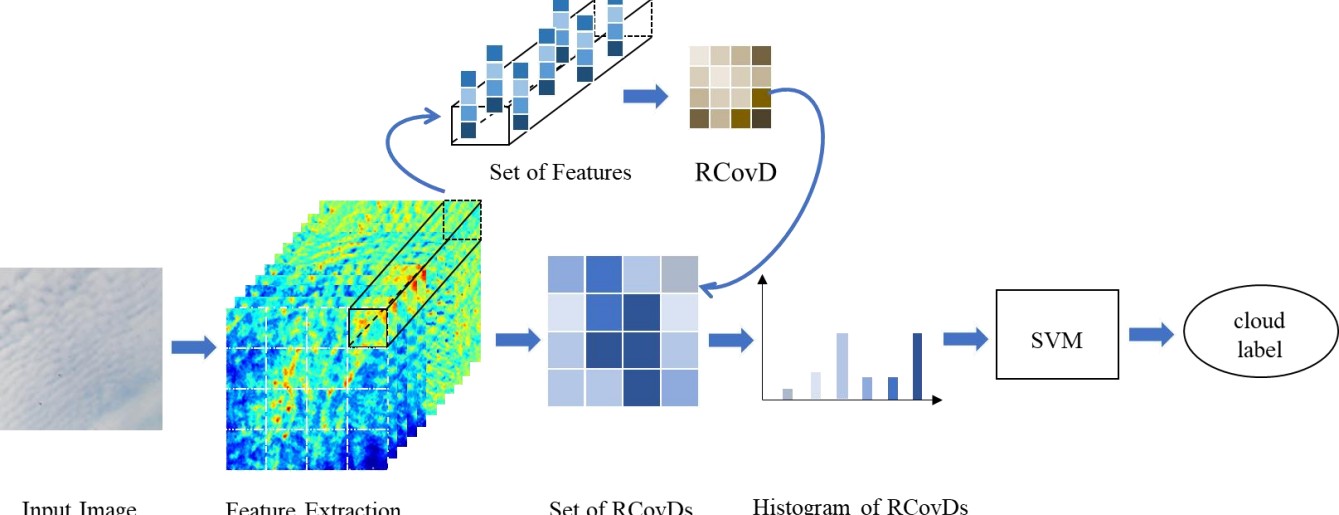

**Figure 1: Pipeline of the proposed cloud classification method. Multiple pixel-level features are firstly extracted from the cloud image blocks to form RCovDs, then the histogram representation of RCovDs are obtained by Riemannian BoF, finally, the cloud type is predicted by multiclass SVM.**



## 2 Data and methodology

### 2.1 Dataset

(1) SWIMCAT dataset: The Singapore Whole sky Imaging CATegories Database (SWIMCAT) was captured by Wide-Angle High-Resolution Sky Imaging System (WAHRISIS) (Dev et al., 2014), a calibrated ground-based whole sky imager. During this observation, from January 2013 to May 2014, different weather conditions spanning several seasons are covered and a far-going cloud categories are collected. The SWIMCAT dataset involves 784 sky/cloud images, including 5 distinct classes: clear sky, patterned clouds, thick-dark clouds, thick-white clouds, and veil clouds. Figure 2 shows sample images from each category, the images have a dimension of $125 \times 125$ pixels (Dev et al., 2015).

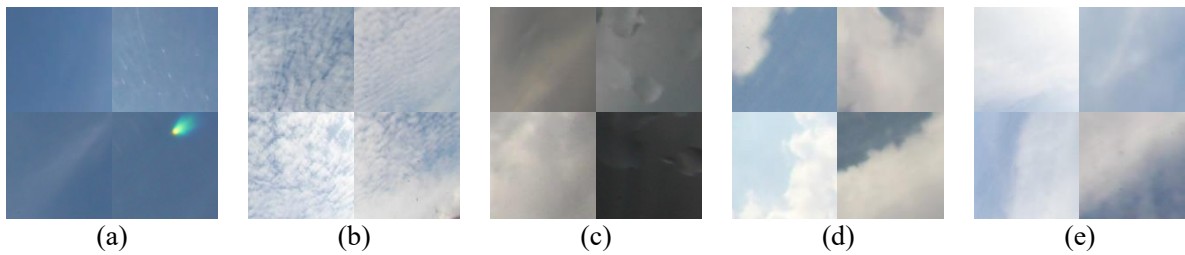

**Figure 2: Sample images from the SWIMCAT dataset. The dataset includes five cloud types, namely, (a) Clear sky, (b) Patterned clouds, (c) Thick-dark clouds, (d) Thick-white clouds, and (e) Veil clouds.**

(2) *zenithal* dataset: This dataset was acquired by the whole-sky infrared cloud-measuring system (WSIRCMS), which is located in Nanjing, China. The *zenithal* dataset contains 500 sky/cloud images, comprising of five different categories: cirriform clouds, clear skies, cumuliform clouds, stratiform clouds and waveform clouds (Liu et al., 2011; Liu et al., 2013). Figure 3 illustrates some sample images of different cloud types, and the image size is $320 \times 240$ pixels.

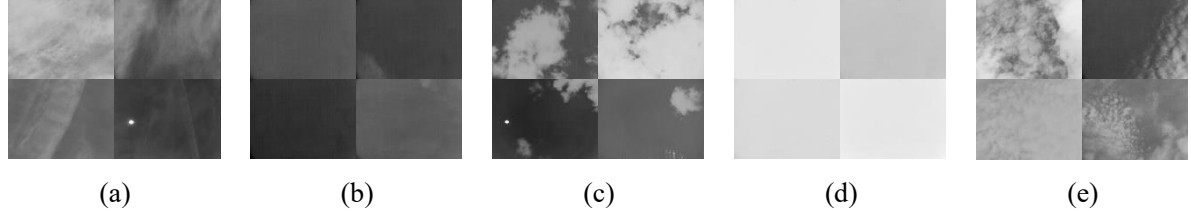

**Figure 3: Sample images from *zenithal* dataset. (a) Cirriform clouds, (b) Clear sky, (c) Cumuliform clouds, (d) Stratiform clouds and (e) Waveform clouds.**

### 2.2 Region Covariance Descriptors

Let $f$ be the $W \times H \times d$ feature map extracted from the cloud image $I$. For a given rectangular region $R$ with size $w \times w$, it contains $n = w \times w$ pixels of $d$-dimensional feature vectors $\{f_i, i = 1, 2 \cdots n\}$. The RCovD is defined by a $d \times d$ symmetric covariance matrix $\mathbf{C}_R$:

$$\mathbf{C}_R = \frac{1}{n-1} \sum_{i=1}^{n} (f_i - \mu)(f_i - \mu)^T \tag{1}$$





where $\mu = \dfrac{1}{n}\sum_{i=1}^{n} f_i$ is the mean of the feature vectors.

The RCovD correlates different components of the feature vectors, the diagonal entry $\mathbf{C}_R(i,i)$ represents the variance of $i$-th components of $n$ feature vectors, and the element $\mathbf{C}_R(i,j)$ denotes the covariance of $i$-th and $j$-th components. Specifically, the RCovD subtracts the mean of the feature vectors, so it can filter out the noise to a certain extent. Note that, there might be a slight chance that $\mathbf{C}_R$ is not strictly positive definite, in this particular case, the $\mathbf{C}_R$ could be converted into a symmetric positive definite (SPD) matrix by adding a regularization term $\lambda E$ , where $\lambda$ is a tiny coefficient which is set to $10^{-4} \times trace(\mathbf{C}_R)$ and $E$ is the identity matrix (Huang et al., 2018; Wang et al., 2012; Wang et al., 2018a).

RCovDs belong to SPD manifold, when it is endowed with a Riemannian metric, it forms a Riemannian manifold. Based on the metric, the geodesic distance can be induced to measure the similarity of the image features. The geodesic distance is the length of the shortest curve between two SPD matrices on SPD Riemannian manifold. The most common distance is the Affine Invariant Riemannian Metric (AIRM) (Pennec et al., 2006):

$$\delta_G(\mathbf{X},\mathbf{Y}) = \left\| \log(\mathbf{X}^{-1/2}\mathbf{Y}\mathbf{X}^{-1/2}) \right\|_F \tag{2}$$

where $\left\| \bullet \right\|_F$ is the Frobenius matrix norm and $\log(\bullet)$ denotes the matrix logarithm. The matrix logarithm can be calculated by singular-value decomposition (SVD), let $\mathbf{A} = \mathbf{U}\Sigma\mathbf{U}^{\mathbf{T}}$ be the eigenvalue decomposition of a symmetric matrix, the logarithm of $\mathbf{A}$ is given by

$$\log(\mathbf{A}) = \mathbf{U}\log(\Sigma)\mathbf{U}^{\mathbf{T}} \tag{3}$$

However, AIRM is computationally demanding. Driven by such computational concerns, in this paper, we adopt the Stein divergence (Sra, 2012) as a Riemannian distance metric, which is defined as

$$\delta_S(\mathbf{X},\mathbf{Y}) = (\log\left|\frac{\mathbf{X}+\mathbf{Y}}{2}\right| - \frac{1}{2}\log\left|\mathbf{X}\mathbf{Y}\right|)^{\frac{1}{2}} \tag{4}$$

where $\left|\bullet\right|$ denotes det operator.

## 2.3 Feature Extraction

The features for cloud type recognition should be representative and discriminative. In this paper, for the *zenithal* dataset, 7 features are extracted, including the image intensity $I(x,y)$, the norms of first and second order derivatives of $I(x,y)$ in both $x$ and $y$ direction, and the norm of gradient. The *zenithal* cloud image is mapped to a 7-dimensional feature space:

$$f_z = \left[ I \quad \left|I_x\right| \quad \left|I_y\right| \quad \left|I_{xx}\right| \quad \left|I_{xy}\right| \quad \left|I_{yy}\right| \quad \sqrt{\left|I_x\right|^2 + \left|I_y\right|^2} \right]^T \tag{5}$$





As for the SWIMCAT dataset, we empirically choose the grayscale of $B$ component, norms of first order derivatives of each color component, and the norm of gradient. Each pixel of the SWIMCAT image is transformed to a 13-dimensional feature map.

$$f_s = [B \ |R_x| \ |R_y| \ |R_z| \ |G_x| \ |G_y| \ |G_z| \ |B_x| \ |B_y| \ |B_z|$$
$$\sqrt{|R_x|^2 + |R_y|^2 + |R_z|^2} \ \sqrt{|G_x|^2 + |G_y|^2 + |G_z|^2} \ \sqrt{|G_x|^2 + |G_y|^2 + |G_z|^2} \ ]^T \tag{6}$$

We divide the cloud image into image blocks and then compute the SPD matrices with the feature maps defined in Eq. (5) and Eq. (6). With the Riemannian BoF, those local feature descriptors in the form of SPD matrices are converted into a histogram feature vector, which is used for cloud type classification.

### 2.4 Riemannian Bag-of-Feature

BoF requires a codebook with $k$ codewords, which are usually obtained by clustering local descriptors. To extend the
conventional BoF from Euclidean space into SPD Riemannian manifold $\mathcal{M}$, two issues should be considered: (1) Construct a codebook $\mathbb{C} = \{\mathbf{C}_j\}_{j=1}^k$ from a set of training RCovDs $\mathbb{X} = \{\mathbf{X}_i\}_{i=1}^M$. (2) Obtain a $k$-dimensional histogram from a set of RCovDs $\mathbb{E} = \{\mathbf{E}_i\}_{i=1}^N$ with the codebook $\mathbb{C}$.

An alternative way to learn a codebook is to apply the conventional $k$-means on vectorized RCovDs in the tangent space (Faraki et al., 2015b), however, it neglects the non-Euclidean geometric structure of SPD matrices. Taking the Riemannian
geometry of SPD matrices into consideration, a possible way is to compute the cluster centers with Karcher mean (Pennec, 2006). The Karcher mean finds a point that minimizes the following object function

$$\mathbf{C}_j^* = \arg \min_{\mathbf{C}_j} \sum_i \delta_S^2 \left( \mathbf{X}_i, \mathbf{C}_j \right) \tag{7}$$

where $\delta_S$ is Stein divergence to measure the geodesic distance of $\mathbf{X}_i$ and the clustering center $\mathbf{C}_j$. Given the training set $\mathbb{X}$, the codebook $\mathbb{C}$ is initialized by randomly selecting $k$ RCovDs from $\mathbb{X}$, and iteratively update the cluster centers using Eq.
(7) until the average distance of each point $\mathbf{X}_i$ to its nearest cluster is minimized. The procedure is summarized in Algorithm 1. We choose the number of codewords empirically by considering the trade-off between classification accuracy and computation consumption, which will be detailed in Section 3.

---

Algorithm 1: ***k*-Means clustering for codebook learning**

---

**Input:**

- training set $\mathbb{X} = \{\mathbf{X}_i\}_{i=1}^M, \ \mathbf{X}_i \in \mathcal{M}$
- $k$, the number of clusters
- *nIter*, the maximum number of iterations

**Output:**

- codebook $\mathbb{C} = \{\mathbf{C}_j\}_{j=1}^k, \ \mathbf{C}_j \in \mathcal{M}$


1: Initialize the codebook $\mathbb{C} = \{\mathbf{C}_j\}_{j=1}^k$ by selecting $k$ samples from $\mathbb{X}$ randomly.

2: **for** $t = 1 \rightarrow nIter$ do

3:   Assign each point $\mathbf{X}_i$ to its nearest cluster in $\mathbb{C}$.

4:   Recompute each cluster center $\mathbf{C}_j^*$ using Karcher mean by minimizing Eq. (7).

5:   Compute the geodesic distance $\varepsilon$ between new cluster center $\mathbf{C}_j^*$ and original cluster center $\mathbf{C}_j$.

6:   If $\varepsilon$ is less than a predefined threshold or $t$ reaches the maximum number of iterations, then break the loop.

7: **end for**

After obtaining the codebook $\mathbb{C}$, the image-level feature can be expressed with the histogram $H$ of RCovDs. In the most straightforward case, $H$ can be yielded by hard assign each RCovD $\mathbf{E}_i$ to the closest codeword in $\mathbb{C}$ with Stein divergence.

The $j$-th ($1 \leq j \leq k$) dimension of $H$ denotes the number of RCovDs assigned to the $j$-th codeword. To demonstrate the significance of histogram feature generated by Riemannian BoF, we randomly select half of images in the SWIMCAT dataset and partition each $125 \times 125$ image into 25 non-overlapping image blocks of size $25 \times 25$ to extract the second-order tensor features in the form of RCovD. Then, we learn a codebook of 10 codewords with Algorithm 1. In the same way, we select 20 images of each cloud type from the remaining images in the SWIMCAT dataset to construct a set of RCovDs for test, and

assign each RCovD to the nearest codeword to obtain the RCovD histogram of each cloud type. As shown in Fig. 4, RCovDs from different cloud types have obviously separable codeword distributions. RCovD distributions of clear sky, pattern and thick-dark clouds are relatively concentrated, while the distributions of thick-white and veil clouds are slightly scattered. In particular, the RCovDs of veil clouds and clear sky are assigned to almost the same codewords, which makes the categorization of these two types challenging. Overall, our proposed Riemannian BoF provides vectorized discriminative representation for

the cloud classification task.

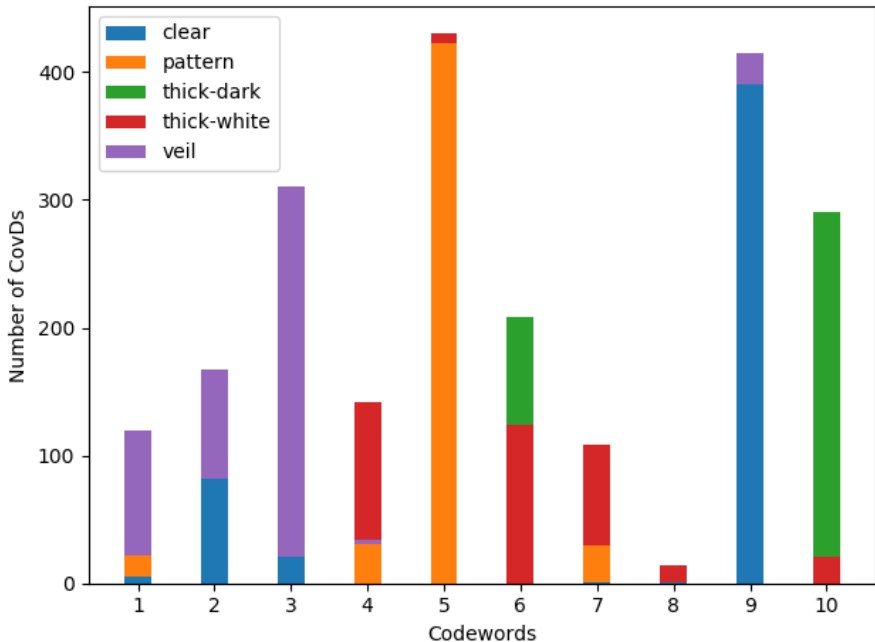

**Figure 4: Histogram of RCovDs from different cloud types on SWIMCAT dataset. RCovDs from different cloud types have distinctive codeword distributions. RCovDs distributions of clear sky, pattern and thick-dark clouds are relatively concentrated, while the distributions of thick-white and veil clouds are slightly scattered. RCovDs of veils clouds and clear sky are assigned to almost the same codewords, which makes the categorization of these two types challenging.**

### 2.5 Classification

SVM has significant performance in the classification task, since it establishes an input-output relationship straightly from the training dataset, and it exclude the need of any priori assumptions or specific preprocessing phases. Another merit is that, once the training procedure is finished, the classification is directly obtained in real time with a strong reduction of computation (Taravat et al., 2015).

For $m$-class classification tasks, there are several ways to build SVM classifiers. In this paper, the "one-against-one" method is adopted, in which $m(m-1)/2$ binary classifiers are constructed, and each classifier distinguishes one cloud type to another. We use the voting strategy to designate the cloud image to the category with the maximum number of votes (Chang and Lin, 2007; Hsu and Lin, 2002; Knerr et al., 1990; Kreßel, 1999). The proposed algorithm is summarized in Algorithm 2, in which SVM is implemented by the LIBSVM toolbox (Chang and Lin, 2007).

Algorithm 2: **The proposed cloud classification algorithm**

**Input:**
- Cloud image $I$ with size $W \times H$
- Codebook $\mathbb{C} = \{\mathbf{C}_j\}_{j=1}^k, \quad \mathbf{C}_j \in \mathcal{M}$





**Output:**

- Cloud type label $L$

1: Extract $W \times H \times d$ feature map $f$ from cloud image $I$

2: Divide $f$ into $N$ blocks with size $w \times w \times d$, and construct RCovDs $\mathbb{E} = \{\mathbf{E}_i\}_{i=1}^{N}$ using Eq. (1)

3: Obtain a $k$-dimensional histogram $H = \{h_j\}_{j=1}^{k}$ representation of $\mathbb{E} = \{\mathbf{E}_i\}_{i=1}^{N}$:

4:    Initialize $H = \{h_j\}_{j=1}^{k}$ with zeros

5:    **for** $i = 1 \rightarrow N$ **do**

6:       Assign $\mathbf{E}_i$ to its nearest codeword $\mathbf{C}_j$, $h_j = h_j + 1$

7:    **end for**

8: Classify $H$ using voting strategy:

9:    Initialize the number of votes $\{V_j\}_{j=0}^{m}$ with zeros

10:    **for** $i = 1 \rightarrow m(m-1)/2$ **do**

11:       Use the $i$-th binary SVM to classify $H$ and obtain the prediction label $L_j$

12:       Update the number of votes by $V_j = V_j + 1$

13:    **end for**

14: Find the maximum number of votes $V_j$ and output the corresponding label $L_j$

## 3 Experiments and discussion

To demonstrate the performance of our proposed cloud type classification method, we conduct several experiments on the SWIMCAT and *zenithal* datasets. We firstly analyze the effects of the two parameters (i.e. the codebook size $k$ and the image block size $w \times w$) involved in the proposed algorithm on cloud type classification accuracy. Then, we design an empirical validation with various training/test partitions. Finally, we quantitatively evaluate and compare the best results of different methods, i.e., WLBP (Liu et al., 2015), BC (Cheng and Yu, 2015), and Luo's methods (Luo et al., 2018).

### 3.1 Parameter Configuration Analysis

In order to assess the impacts of the codebook size, i.e., the centroids number $k$, and the image block size $w \times w$ on cloud classification accuracy, we conduct sensitivity analysis on the SWIMCAT and *zenithal* datasets. In our experiments, $k$ ranges from 5 to 40 with interval 5 and $w$ ranges from 8 to 120 with the step size of 4. For a given $w$, the $W \times H \times d$ feature map is divided into $\lfloor \frac{W}{w} \rfloor \times \lfloor \frac{H}{w} \rfloor \times d$ blocks start from the upper left corner of the feature map and the incomplete blocks at the edges are dropped. We randomly choose 9/10 images of the dataset for training and the rest is for testing. The classification accuracy of each parameter configuration, as shown in Fig. 5, indicates that, to a certain extent, the larger the number of codebook size,



the better performance on both datasets. However, we observe that the improvement is not statistical significance after $k$
exceeds 20, while the computing burden increases obviously. In fact, the complexity of the Riemannian BoF is mainly
determined by the cluster center number. We note that as the block size $w$ increases, the classification accuracy increases first
and then degrades beyond the highest point, this trend is especially evident on *zenithal* dataset. The reason is that larger blocks
can capture more abundant texture information, while the local details might be ignored. Therefore, in the following
experiments, considering trade-offs between classification accuracy and efficiency, we set $k = 30$, $w = 24$ for the SWIMCAT
dataset, and $k$ and $w$ are set to 35 and 52 for the *zenithal* dataset.

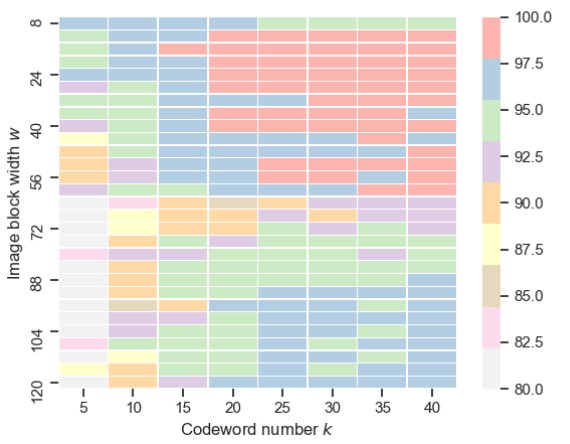 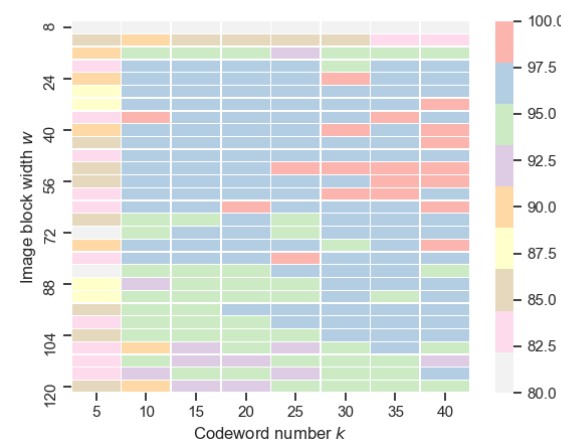

(a) Sensitivity analysis on the SWIMCAT dataset      (b) Sensitivity analysis on the *zenithal* dataset

**Figure 5: Sensitivity analysis of parameter $k$ and $w$ in the proposed method on the SWIMCAT dataset and *zenithal* dataset.**

### 3.2 Evaluation on Dataset with Small Sample Size

In machine learning tasks, suitable annotated data samples are in short supply and quite costly for classifier training and testing.
Since manual labeling requires much workforce, it is of great significance to reduce the dependence of the classification model
on the labeled dataset. To estimate the performance of the proposed method comprehensively, we extract different proportions
of training images randomly from each dataset and take the rest images as the test set. In order to guarantee the stability of the
classification results, each experiment was repeated five times to take the average as the final classification result. Figure 6
shows that in the situation of small sample size, for the SWIMCAT dataset, the proposed method achieves accuracy more than
90% on the test set with only 3% images (i.e., 24/784) of the dataset as the training set. The accuracy can be improved by 5%
at least when the training set accounts for 9% images (i.e., 72/784). As for the *zenithal* dataset, our method obtains more than
90% classification accuracy on the test set when we randomly select 6% images (i.e., 30/500) of the dataset as training set, and
achieving more than 95% accuracy when the proportion of training images increases to 10%. Generally, our proposed method
significantly fulfills a high classification accuracy in small training sample situations. This is remarkable, considering that our
proposed method is combining just RCovDs and Riemannian BoF. In conclusion, the proposed method requires only a few
manually labeled samples to achieve a high cloud type recognition accuracy.





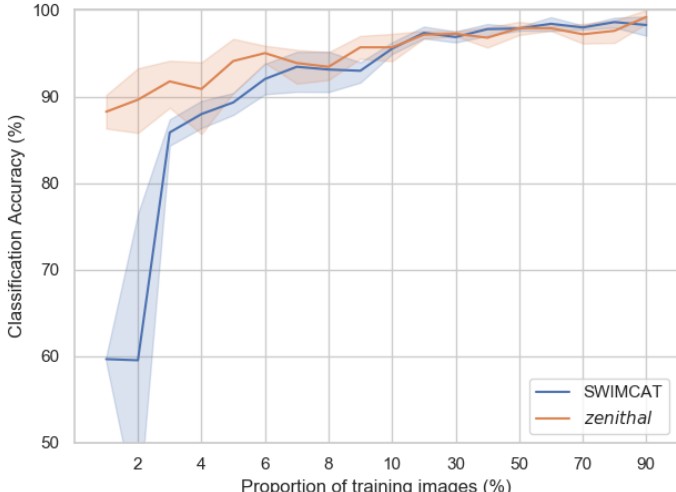

**Figure 6: Performance analysis of training images with different proportions on the SWIMCAT dataset and *zenithal* dataset.**

### 3.3 Comparison with state-of-the-art methods

Iterated cross validation is chosen as an effective scheme to verify the performance of the classifier. This strategy estimates

the performance by randomly choose a part of the samples for independent training and testing the model without these samples, and repeating the procedure dozens of times (Beleites et al., 2013). In each experiment, we randomly select the same proportion (i.e., 1/10, 5/10, 9/10) of images for each category as the training set, and the remaining images are used as the test set. Each classification experiment is repeated 50 times to obtain the average accuracy as the final experimental result.

We compare the performance of our method with the best results published on the SWIMCAT dataset in Table 1.Notice

that our algorithm utilizing RCovDs has a 2.58% accuracy rate at SWIMCAT dataset than other methods when the training sample accounts for 1/10 of the total data. And when the training sample accounts for 5/10 and 9/10, the proposed method is slightly higher than Luo's method and much higher than the other two methods. Figure 7 shows the confusion matrix of classification results with our proposed method on SWIMCAT dataset, with 9/10 of the dataset as training set. The discrimination rates of clear sky, pattern clouds and thick-dark clouds are perfect 100%, which demonstrates that these three

types tend to be easily distinguished among all cloud types since they have the most significant features. Figure 8 shows two misclassified examples of SWIMCAT dataset, where yellow labels are the ground truth, and the red labels are the predicted cloud types predicted by our method. Notice that the veil clouds are prone to be misclassified as clear sky, since the veil clouds are thin and have highlight transmittance. Moreover, some veil clouds are misclassified as thick-white cloud, when the camera lens is contaminated, and the clouds is too thick. Besides, a small amount of thick-white clouds is misclassified as clear sky,

pattern clouds or veil clouds.

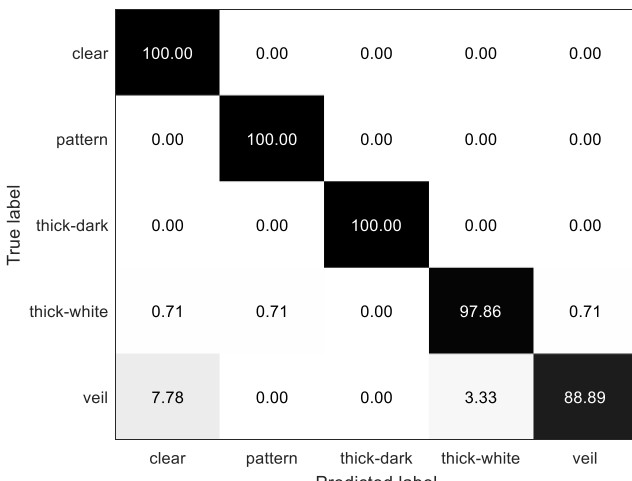

**Figure 7: The confusion matrix of the SWIMCAT dataset classification results using our proposed method. 9/10 of the dataset is used for training and the rest is used for testing, the overall classification accuracy is 98.4%.**

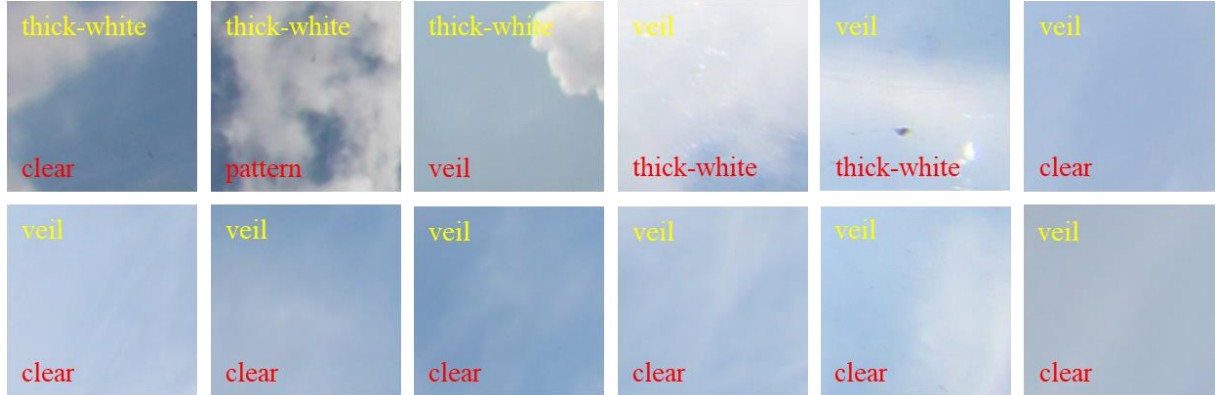

**Figure 8: Misclassified images of SWMINCAT dataset. The yellow labels are the ground truth, and red labels are predicted cloud types. The veil clouds are prone to be misclassified as clear sky, since the veil clouds are thin and have high light transmittance, some veil clouds are misclassified as thick-white cloud, when the camera lens is contaminated and the clouds is too thick. Besides, a small amount of thick-white clouds is misclassified as clear sky, pattern clouds or veil clouds.**

As for the *zenithal* dataset, Table 2 illustrates that the proposed method gains the highest overall accuracy compared with the other approaches. Figure 9 displays the confusion matrix of classification results with our method on the *zenithal* dataset, when 90% of the dataset is used as the training set. The discrimination rates of clear sky, cumuliform clouds and stratiform clouds are up to 100%. Only a small part of waveform clouds is misclassified as clear sky or cirriform clouds. In addition, some of the cirriform clouds are misclassified as clear sky or waveform clouds. Figure 10 illustrates the misclassified images of the *zenithal* dataset, waveform clouds and cirriform clouds are easy to be categorized as clear sky if the size of sky area is much larger than that of clouds. The reason why the waveform clouds and cirriform clouds are confused with each other is that they sometimes own extremely similar textures.





**Table 1: Classification accuracy (%) of the SWIMCAT dataset obtained by different methods.**

| Method | 1/10 | 5/10 | 9/10 |
|---|---|---|---|
| WLBP**(Liu et al., 2015)** | 72.31 | 84.52 | 88.86 |
| BC**(Cheng and Yu, 2015)** | 93.86 | 94.87 | 95.04 |
| LUO**(Luo et al., 2018)** | 91.83 | 97.72 | 97.86 |
| Our method | **96.44** | **98.40** | **98.40** |

**Table 2: Classification accuracy (%) of the *zenithal* dataset obtained by different methods.**

| Method | 1/10 | 5/10 | 9/10 |
|---|---|---|---|
| WLBP**(Liu et al., 2015)** | 81.64 | 92.24 | 93.48 |
| BC**(Cheng and Yu, 2015)** | 81.30 | 81.32 | 81.32 |
| LUO**(Luo et al., 2018)** | 90.85 | 95.98 | 96.36 |
| Our method | **95.00** | **97.40** | **98.60** |

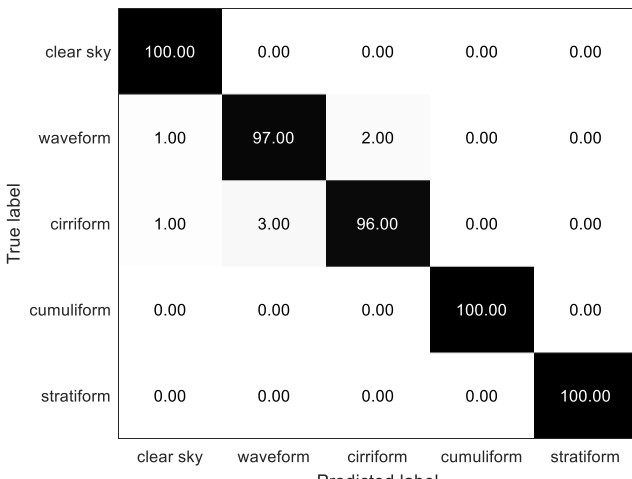

**Figure 9: The confusion matrix of the *zenithal* dataset classification results using the proposed method. 9/10 of the dataset is used for training and the rest is used for testing, the overall accuracy is 98.6%.**





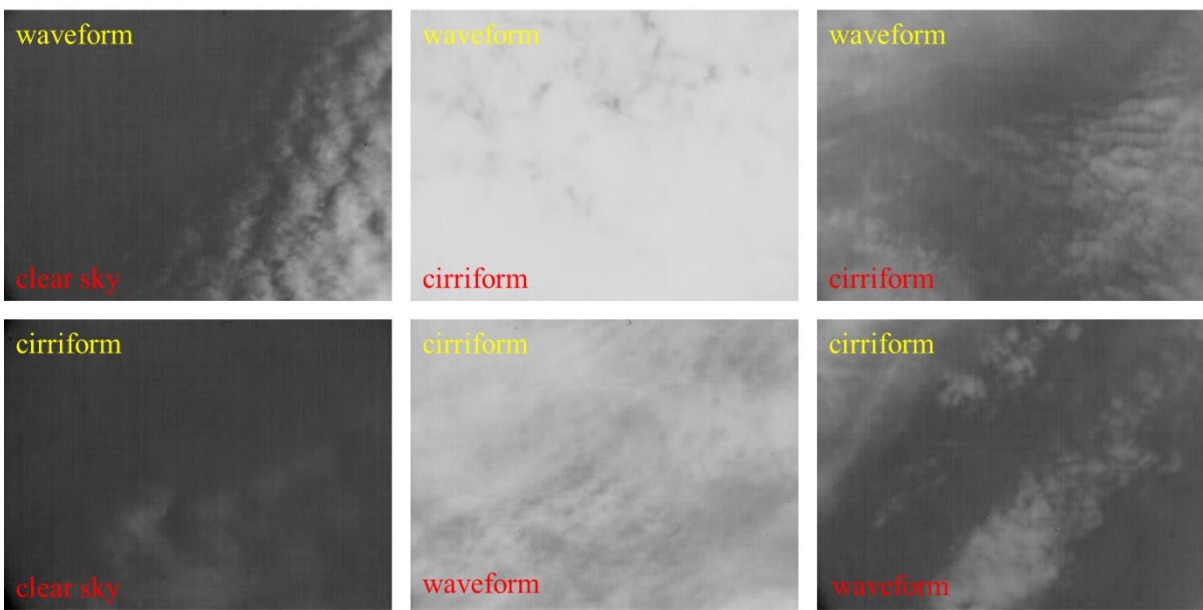

**Figure 10: Misclassified images of the *zenithal* dataset. The yellow labels are the ground truth, and red labels are predicted cloud types. Waveform clouds and cirriform clouds are categorized as clear sky because the size of sky area is much larger than that of clouds, and these two cloud types are easily confused as they share similar local patterns.**

## 4 Conclusions

To tackle the challenge of automatic cloud type classification for ground-based cloud images, in this paper, we present a new classification method with RCovDs as the local feature representation. RCovDs provide a simple way to fuse multiple pixel-level features, which improves discriminative ability for cloud images. The image-level information is obtained by applying Riemannian BoF to encode RCovDs into a histogram. Finally, we apply the "one-against-one" multi-class SVM as the classifier.

It is noted that even we choose relatively simple image features to calculate RCovDs, the performance of the proposed method is still impressive. We conduct parameter analysis experiment and figure out how block size $w$ and codewords number $k$ affect the accuracy of the proposed method. Classification experiments with different training set sizes demonstrate that our method is still efficient in the case of small size training set, which can greatly reduce the labor for labeling. In the third experiment, we compare our method to the other three cloud classification algorithms with different configurations of training/test sets. As the experimental results validate, the proposed method is competitive to state-of-the-art methods on both SWIMCAT and *zenithal* datasets.

In future work, the features like LBP or GLCM could be gathered and mapped into Riemannian manifold and multi-scale block strategy can be taken into consideration for a higher cloud type categorization accuracy. Others, the complex sky conditions with various cloud types should be deeply investigated to fulfill the application needs.



*Code and data availability*. The code of the proposed method can be made available via email to tangyuzhu9293@163.com. The SWIMCAT dataset used in this paper is available for download from http://vintage.winklerbros.net/swimcat.html, and the *zenithal* dataset can be made available via email to tangyuzhu9293@163.com.


*Author contributions*. YT performed the experiments and wrote the paper. PY analysed the data and designed the experiments. ZZ conceived the method and reviewed the paper. JC, DP and XZ reviewed the paper and gave constructive suggestions.

*Competing interests*. The authors declare that they have no conflict of interest.


*Acknowledgements*. This work is supported by the National Key Research and Development Program of China (Grant No. 2016YFC1400903), NSFC-Zhejiang Joint Fund for the Integration of Industrialization and Informatization (Grant No. U1609202) and the National Natural Science Foundation of China (Grant Nos. 61473310, 41174164, 41775027, 41376184 and 40976109).

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
