# Peer review of "Improving Cloud Type Classification of Ground-Based Images Using Region Covariance Descriptors"

_Atmospheric Measurement Techniques, 2020_

## Referee Comment (RC1) · Anonymous Referee #2 · 30 Aug 2020

Summary:

Apply region covariance descriptor (RCovD) computations to features and then use the output from the RCovD computations as input to a Bag-of-Features approach to create histograms. The histograms are classified using a support vector machine (SVN) method. The approach is applied to labelled cloud images, with some images serving as training data and the remaining images as testing data. Results of the classification are given.

Overall Comments:

From an atmospheric science perspective, a weakness in the paper in regards to its

publication in AMT is its weak link to atmospheric science applications. The Abstract contains the sentences

"Cloud types are important indicators of cloud characteristics and short-term weather forecasting. The meteorological researchers can benefit from the automatic cloud type recognition of massive images captured by the ground-based imagers. However, by far it is still of huge challenge to design a powerful discriminative classifier for cloud categorization."

and the first few sentences of the Introduction are as follows:

"Clouds have a strong impact on climate modeling, weather prediction and the Earth's energy budget balance. In recent years, the growing appeal on renewable solar energy pushes additional interest on cloud coverage measurement and cloud classification (Heinemann et al., 2006; J. Huertas, 2017; Martínez-Chico et al., 2011). Therefore, accurate cloud type classification is in great need."

Just how cloud classification as pursued in this study is applicable to climate modeling, weather prediction, Earth's energy balance, and surface solar irradiance is never made. Advancing our knowledge on these topics requires three-dimensional fields of optical depth, water content, and hydrometeor numbers, shapes, sizes, phase, and fall speeds. How results of the classification scheme have a bearing on these quantities is never made. Related to this issue is that the physical significance of the five cloud types in the 784 images of the SWIMCAT dataset and the 500 images of the Zenithal dataset is never made. For example, what is the physical significance of "patterned clouds, thick-dark clouds, thick-white clouds" and how will knowledge of their occurrence provide information on improving climate and forecast models and studies? Not clear.

This study is similar to those that occurred in the late 1980s and 1990s during the first wave of artificial intelligence/machine learning methods into the atmospheric sciences. Many studies were devoted to classifying cloud types, cloud textures, etc..., similar to

the current study, but these studies never really went anywhere because of lack of quantitative information relative to cloud optical depths, water contents, etc... How this paper will escape this same fate is not addressed.

Perhaps the proposed approach of combining RCovDs, Bag-of-Feature, and SVMs for classification has value relative to existing techniques. To assess this possibility, it would be more convincing to apply the proposed algorithm to large, vetted datasets in the artificial intelligence/machine learning community and to have this community rigorously assess its results. Along this same line of reasoning, it would strengthen the paper if a case could be made as to why the results presented in this paper using only 784 and 500 images are "impressive" as stated on Line 259 of the paper. Because not much was stated about the diversity of clouds in these few scenes, it is hard to tell if high classification accuracies in regards to them are compelling.

Overall Weaknesses:

The relevance of the paper to outstanding and important issues in weather and climate is not made in a compelling fashion.

Along this same line, the output of the classification algorithm does not contain quantitative information on fundamental hydrometeor properties. Rather, it provides classifications of cloud patterns whose fundamental importance are unknown. As a result, the significance of the results to weather and climate problems is not clear.

Finally, the datasets used in the study are relatively small and of unknown diversity and significance. From a purely algorithmic development perspective, testing it on accepted datasets for algorithm evaluation would be much more compelling.

---

## Author Comment (AC1) · 1 Sep 2020

**Response to Reviewer#2**

Sincerest thanks for the comments on our manuscript entitled "Improving Cloud Type Classification of Ground-Based Images Using Region Covariance Descriptors" (amt-2020-189). These comments are very valuable and helpful for revising and improving our paper, and they also have important guiding significance to our researches. We have studied comments carefully and made revisions which we hope meet with approval.

The main revisions in the paper and the responses to the reviewer's comments are as following:

**Reviewer #2:**

*Summary:*

*Apply region covariance descriptor (RCovD) computations to features and then use the output from the RCovD computations as input to a Bag-of-Features approach to create histograms. The histograms are classified using a support vector machine (SVM) method. The approach is applied to labelled cloud images, with some images serving as training data and the remaining images as testing data. Results of the classification are given.*

*1. The Abstract contains the sentences "Cloud types are important indicators of cloud characteristics and short-term weather forecasting. The meteorological researchers can benefit from the automatic cloud type recognition of massive images captured by the ground-based imagers. However, by far it is still of huge challenge to design a powerful discriminative classifier for cloud categorization." and the first few sentences of the Introduction are as follows: "Clouds have a strong impact on climate modeling, weather prediction and the Earth's energy budget balance. In recent years, the growing appeal on renewable solar energy pushes additional interest on cloud coverage measurement and cloud classification (Heinemann et al., 2006; J. Huertas, 2017; Martínez-Chico et al., 2011). Therefore, accurate cloud type classification is in great need."*
*Just how cloud classification as pursued in this study is applicable to climate modeling, weather prediction, Earth's energy balance, and surface solar irradiance is never made. Advancing our knowledge on these topics requires three-dimensional fields of optical depth, water content, and hydrometeor numbers, shapes, sizes, phase, and fall speeds. How results of the classification scheme have a bearing on these quantities is never made. Related to this issue is that the physical significance of the five cloud types in the 784 images of the SWIMCAT dataset and the 500 images of the Zenithal dataset is never made. For example, what is the physical significance of "patterned clouds, thick-dark clouds, thick-white clouds" and how will knowledge of their occurrence provide information on improving climate and forecast models and studies? Not clear.*

Reply: Many thanks for your concern. Motivated by the comments, we have detailed the impact of cloud classification on climate modeling, weather prediction, Earth's energy balance, and surface solar irradiance in the revised manuscript. The sentences at the beginning of the Introduction section have been replaced as following:

"Clouds have a strong impact on the Earth's energy budget balance, climate modeling and weather prediction. Cloud type variations (e.g., variations in cloud-top height and water content) may affect both shortwave and longwave radiative fluxes. During climate variations, the distribution and frequency of different cloud types may change (Chen et al., 2000). Additionally, accurate cloud classification, especially convective cloud identification, are essential to Hazardous weather monitoring (Zhang et al., 2018a)."

In this paper, we extend our previous work[1], which has been published in AMT. Currently, ground-based cloud observation equipment can only estimate cloud types and cloud cover, other cloud characteristics like optical depth, water content, and hydrometeor numbers, shapes, sizes, phase, and fall speeds, etc., need to be acquired by other detection equipment, which is not the focus of this paper. The main contribution of our work is to propose a new automatic cloud type classification method with better performance as compared to state-of-the-art approaches, especially for the small training dataset. The physical significance of cloud type classification results on improving climate and forecast models may beyond the scope of our current paper, and we will explore this topic in the future work.

[1] Luo, Q., Meng, Y., Liu, L., Zhao, X., and Zhou, Z.: Cloud classification of ground-based infrared images combining manifold and texture features, Atmos. Meas. Tech., 11, 5351-5361, 10.5194/amt-11-5351-2018, 2018.

*2. This study is similar to those that occurred in the late 1980s and 1990s during the first wave of artificial intelligence/machine learning methods into the atmospheric sciences. Many studies were devoted to classifying cloud types, cloud textures, etc..., similar to the current study, but these studies never really went anywhere because of lack of quantitative information relative to cloud optical depths, water contents, etc... How this paper will escape this same fate is not addressed.*

Reply: We appreciate for your valuable comment. In this manuscript, we provide an improved ground-based cloud images classification method with region covariance descriptors (RCovDs) and Riemannian Bag-of-Feature (BoF). The main perspective of this paper is to solve the problem of cloud categorization automatically with the aid of computer visual technique. Perhaps carrying out deep research on physical properties of clouds might help us understanding the structure of clouds, further help us extracting discriminative features and accomplishing cloud classification tasks. And we will explore this topic in the future work.

*3.Perhaps the proposed approach of combining RCovDs, Bag-of-Feature, and SVMs for classification has value relative to existing techniques. To assess this possibility, it would be more convincing to apply the proposed algorithm to large, vetted datasets in the artificial intelligence/machine learning community and to have this community rigorously assess its results.*

*Along this same line of reasoning, it would strengthen the paper if a case could be made as to why the results presented in this paper using only 784 and 500 images are "impressive" as stated on Line 259 of the paper. Because not much was stated about the diversity of clouds in these few scenes, it is hard to tell if high classification accuracy in regard to them are compelling.*

Reply: Thanks for your comment. Unlike the deep learning-based models (e.g., VGGNet, LeNet and ResNet) that designed for general pattern recognition tasks, the proposed algorithm is specifically designed for ground-based cloud image categorization. Although image classification frameworks based on deep learning have achieved huge success, these frameworks all rely on a big pretrained model based on a large-scaled related data set in which the images are finely labelled. However, there is no appropriate public data set large enough to train a deep learning model for cloud classification. In fact, constructing a large data set, including multiple cloud types and finely labelled the cloud images, is much more difficult and expensive because annotating the cloud images needs professional experts with rich observing experience. The advantage of our model is that a very high prediction accuracy can be obtained with a small number of training samples, which is demonstrated in Section 3.2. Furthermore, the proposed method is tested on two datasets captured by different devices (SWIMCAT dataset and *zenithal* dataset). The SWIMCAT dataset is published by the Vision & InterAction Group (part of the School of Computing at the National University of Singapore) and has now been used as benchmark for cloud classification by most researchers. The *zenithal* dataset published by our team is also available to community. In order to exhibit the diversity of clouds, we showed more sample images in Fig. 2 and Fig.3 in the latest manuscript.

*4. Overall Weaknesses: The relevance of the paper to outstanding and important issues in weather and climate is not made in a compelling fashion. Along this same line, the output of the classification algorithm does not contain quantitative information on fundamental hydrometeor properties. Rather, it provides classifications of cloud patterns whose fundamental importance are unknown. As a result, the significance of the results to weather and climate problems is not clear. Finally, the datasets used in the study are relatively small and of unknown diversity and significance. From a purely algorithmic development perspective, testing it on accepted datasets for algorithm evaluation would be much more compelling.*

Reply: Many thanks for your overall comments. Cloud characteristics like quantitative information and hydrometeor properties need to be acquired by other detection equipment, which is not the focus of this paper. In the last decade, dealing with the problem of cloud classification automatically from the perspective of computer vision has gradually become a trend. A few researchers have been engaged in this field, and almost of them have emphasized the importance of cloud classification for cloud observation [2, 3]. Thus, we believe the categorization of cloud patterns is quite crucial. This manuscript mainly focuses on the cloud image classification algorithm, so the way how the cloud types affect climate and weather is not described in detail. And the reply about the datasets has already been illustrated clearly in the former reply.

[2] Chen, T., Rossow, W. B., and Zhang, Y.: Radiative Effects of Cloud-Type Variations, J. Clim., 13, 264-286, 10.1175/1520-0442(2000)013<0264:reoctv>2.0.co;2, 2000.

[3] Zhang, J., Pu, L., Zhang, F., and Song, Q.: CloudNet: Ground-Based Cloud Classification With Deep Convolutional Neural

90    Network, Geophys. Res. Lett., doi: 10.1029/2018GL077787, 2018a. 10.1029/2018GL077787, 2018

**Once again, thank you very much for your creative comments and suggestions.**

---

## Referee Comment (RC2) · Anonymous Referee #3 · 18 Nov 2020

The manuscript appears to present evidence that using special kind of localised feature extraction in images, together with a specialised measure of similarity between these features, it is possible to train a support vector machine to pick between a set of cloud types that the authors have defined.

Unfortunately the manuscript has structural issues, unclear formulations and lacks details in parts which hampers its ability to communicate the research to the reader. In addition the manuscript lacks evidence of the physical relevance of the particular cloud types used - how specifically are these cloud types important for climate predictions and weather forecasting?

[Figure]

The performance analysis is a bit weak as the datasets used are quite small in size (784 for SWIMCAT and 500 zenithal). I would suggest using data-augmentation (random rotations and zoom) to create two orders of magnitude more images to work with. In addition, because random subsets of the datasets are used for training and validation it is unclear to me whether the performance difference to prior work is actually due to random chance (training/testing on easier partition if the datasets) or whether the technique presented here is indeed better. It is common in machine learning datasets to train all models on the same training set and validate against the same validation set (see for example CIFAR http://www.cs.toronto.edu/∼kriz/cifar.html and MNIST http://yann.lecun.com/exdb/mnist/).

In addition there are structural issues in the manuscript, for example many sentences combine terms that are not on equal footing and often it is not detailed how and why specific assertions are evidenced. For example in the introduction's first sentence: it is true that "clouds have a strong impact" on "Earth's energy budget", but it isn't clear how clouds having a "strong impact" on "climate modelling" or "weather prediction". Further, line 24 goes on to talk about "weather monitoring" rather than "modelling" which are not equivalent. Finally, a extensive list of publications covering "cloud coverage measurement" and "cloud classification", but this publication doesn't appear to be about "cloud coverage". The fact that there is "additional interest" doesn't evidence that "cloud type classification" is in great need, or is that in fact what these papers state?

Specific comments:

Abstract:

l 10: "Cloud types are important indicators of ... short-term weather forecasting" - this sentence doesn't make sense. "Cloud types" can't "indicate" "weather forecasting"

l 11: "The meteorological researchers can benefit from the automatic cloud type recognition of massive images captured by the ground-based imagers". Why is this true? Also, I would leave out "The" in "The meteorological" and the word "massive".

l 12: "However, by far it is still of huge challenge to design a powerful discriminative classifier for cloud categorization" - why is it a huge challenge?

l 14: "BoF is extended from Euclidean space to Riemannian manifold by k-Means clustering, in which Stein divergence is adopted as a similarity metric" - what is the relevance of this? Why is this done?

l 15: "The histogram feature is extracted by encoding RCovDs of the cloud image blocks with BoF-based codebook" - the term "histogram feature" hasn't been explained yet, what is this? How is it relevant to the technique/results of this paper?

l 17: "The experiments on the ground-based cloud image datasets validate the proposed method and exhibit the competitive performance against state-of-the-art methods." - this should instead specify exactly what the improvements on previous work is, give the numbers that indicate the improvement and what the implications of these improvements are.

General:

- the section on "Feature extraction" should be before "Region Covariance Descriptors" since the covariance descriptors used the features.

l 102: the relationship between w in the "Rectangular region R with size w x w" and the width of the input image isn't specified.

I would be happy to review this article again once the above issues have been addressed and a general read through considering the "how" and "why" of each sentence are detailed. The technique presented is interesting and the results encouraging, but the presentation needs improving, details need adding and comparison to prior work could benefit from more comprehensive analysis.

—————————————————————

---

## Author Comment (AC2) · 10 Dec 2020

**Response to Reviewer#3**

Sincerest thanks for the comments on our manuscript entitled "Improving Cloud Type Classification of Ground-Based Images Using Region Covariance Descriptors" (amt-2020-189). These comments are very valuable and helpful for revising and improving our paper, and they also have important guiding significance to our research. We have studied comments carefully and made revisions which we hope meet with approval.

The main revisions in the paper and the responses to the reviewer's comments are as following:

**Reviewer #3:**

*The manuscript has structural issues, unclear formulations and lacks details in parts which hampers its ability to communicate the research to the reader. The manuscript lacks evidence of the physical relevance of the particular cloud types used - how specifically are these cloud types important for climate predictions and weather forecasting? In addition, there are structural issues in the manuscript, for example many sentences combine terms that are not on equal footing and often it is not detailed how and why specific assertions are evidenced. For example, in the introduction's first sentence: it is true that "clouds have a strong impact" on "Earth's energy budget", but it isn't clear how clouds having a "strong impact" on "climate modelling" or "weather prediction". Further, line 24 goes on to talk about "weather monitoring" rather than "modelling" which are not equivalent. Finally, an extensive list of publications covering "cloud coverage measurement" and "cloud classification", but this publication doesn't appear to be about "cloud coverage". The fact that there is "additional interest" doesn't evidence that "cloud type classification" is in great need, or is that in fact what these papers state?*

Reply: Many thanks for the comment and we have added the content of how cloud types affect climate change in Introduction Section of the revised manuscript. Introduction is modified as follows:

"Clouds affect the Earth's climate by modulating Earth's basic radiation balance (Hartmann et al., 1992; Ramanathan et al., 1989). Cloud type variations are shown to be as important as cloud cover in modifying the radiation field of the earth–atmosphere system. For example, stratocumulus, altostratus, and cirrostratus clouds produce the largest annual mean changes of the global top-of-atmosphere and surface shortwave radiative fluxes (Chen et al., 2000). Cloud type is also one of the most reliable predictors of weather, e.g., cirrocumulus clouds are a sign of good weather. Therefore, accurate cloud type classification is in great need. Currently, the classification task is mainly undertaken by manual observation, which is labour-intensive and time-consuming. Benefiting from the development of ground-based cloud image devices, we are able to continuously acquire cloud images and automatically classify the cloud types."

*The performance analysis is a bit weak as the datasets used are quite small in size (784 for SWIMCAT and 500 zenithal). I would suggest using data-augmentation (random rotations and zoom) to create two orders of magnitude more images to work with.*

Reply: Thanks for your concern. Data-augmentation is a great way to expand a limited dataset, especially for deep learning, however, it's not a magic bullet. One always runs the risk of overfitting the data model to the training samples if relying too much on data augmentation technique. If the dataset size expands to create two orders of magnitude more images, the distribution of the sample features may make the model deviate from the real data.

To validate the assumption, we conduct an experiment, in which we compare the predict accuracy of our model trained by data with/without augmentation. Let M1 and M2 denote the proposed model trained without and with data augmentation, respectively. In the training stage of M2, we randomly rotate and flip each training image to generate 9 more images, as shown in Fig. 1 and Fig. 2. The average accuracy obtained by 10-fold cross validation is reported in Table 1. The experiment result shows that, compared to M2, M1 acquires higher recognition accuracy on both datasets. The reason is that the RCovDs used in our model owns rotation and scale invariance characteristics. Moreover, the BoF method makes no use of the location information of RCovDs. Therefore, based on the experimental result, it seems that the data augmentation technique does not significantly improve the accuracy of the proposed model. Others, as mentioned in the paper, one purpose of our model is to address the dilemma that how we can obtain a high cloud type recognition accuracy even with a limited training set.

Table 1. The accuracy of M1 and M2 test on SWIMCAT and *zenithal* dataset.

|  | Accuracy of SWIMCAT dataset | Accuracy of *zenithal* dataset |
| --- | --- | --- |
| Model trained without data augmentation (M1) | **98.4%** | **98.6%** |
| Model trained with data augmentation (M2) | 97.7% | 94% |

[Figure]

Figure 1 Samples of SWIMCAT data augmentation. The upper left image is the original image, and the other images are generated by random flip and rotation.

[Figure]

Figure 2 Sample of *zenithal* data augmentation. The upper left image is the original image, and the other images are generated by random flip and rotation.

*In addition, because random subsets of the datasets are used for training and validation it is unclear to me whether the performance difference to prior work is actually due to random chance (training/testing on easier partition if the datasets) or whether the technique presented here is indeed better. It is common in machine learning datasets to train all models on the same training set and validate against the same validation set.*

Reply:To evaluate the performance of machine learning model, we need to test it on unknown data (or first seen data). Cross validation is one of the techniques used to test the effectiveness of a machine learning model. To perform cross validation, we need to keep aside a sample/portion of the data on which is not used to train the model, later use this sample for testing/validating. The commonly used cross validation techniques are ***holdout method*** and ***k-fold cross validation***.

In the holdout method, we randomly split complete data into training and test sets, ideally split the data into 70%:30% or 80%:20%. Then perform the model training on the training set and use the test set for validation purpose. In typical cross-validation, results of multiple runs of model-testing are averaged together; in contrast, the holdout method, in isolation, involves a single run. The disadvantage of the holdout method is that the performance evaluation is sensitive to the split ratio of the training set and the validation set. It should be used with caution because without such averaging of multiple runs, one may achieve highly misleading results. If our data is huge and our test samples and train samples have the same distribution, then this approach is acceptable. Therefore, this method is usually used in deep learning models with **large scale dataset**.

In $k$-fold cross validation is a popular technique that generally results in a less biased model. Because it ensures that every observation from the original dataset has the chance of appearing in training and test set. This is one among the best approach if we have a **limited input data**. This method follows the below steps: (1) Split the entire data randomly into $k$ folds. (2) Then train the model using the $k$-1 folds and validate the model using the remaining $k$-th fold. (3) Repeat this process until every $k$ - fold serve as the test set. Then take the average of the recorded scores. That will be the performance metric for the model.

In our experiment, as the data size is limited, we choose $k$-fold cross validation for model validating. All comparison methods are conducted under the exact same training/test set partition during each k-fold cross validation process. There won't be a convincing problem since we repeat 50 times k-fold cross validation and take the average accuracy as final result.

*Specific comments:*

*Abstract:*

*l 10: "Cloud types are important indicators of ... short-term weather forecasting" – this sentence doesn't make sense. "Cloud types" can't "indicate" "weather forecasting. "The meteorological researchers can benefit from the automatic cloud type recognition of massive images captured by the ground-based imagers". Why is this true? Also, I would leave out "The" in "The meteorological" and the word "massive".*

Reply: Thanks for your concern. Currently, the classification task is mainly undertaken by manual observation, which is labour-intensive and time-consuming. Benefiting from the development of ground-based cloud image devices, we are able to continuously acquire cloud images and *automatically* classify the cloud types. We will delete the first two sentences "Cloud types are important indicators of ... captured by the ground-based imagers.". Following sentences will be added at the beginning of the Abstract.

"The distribution and frequency of occurrence of different cloud types affects the energy balance of the earth. Automatic cloud type classification of images continuously observed by the ground-based imagers cloud help climate researchers uncover the relationship between cloud type variations and climate change."

*l 12: "However, by far it is still of huge challenge to design a powerful discriminative classifier for cloud categorization" - why is it a huge challenge?*

Reply: Thanks for your comment. The challenges are explained in Introduction Section. The main challenges of the ground-based cloud image classification task can be ascribed to the following reasons: (1) One single feature cannot effectively describe different types of clouds, we need to extract textural, structural, and statistical features simultaneously. (2) The scale of cloud varies greatly, therefore, the extracted features should be robust in the presence of illumination changes and nonrigid motion. (3) Different cloud types may have similar local characteristics, and thus the global features need to be considered.

*l 14: "BoF is extended from Euclidean space to Riemannian manifold by k-Means clustering, in which Stein divergence is adopted as a similarity metric" - what is the relevance of this? Why is this done?*

Reply: RCovDs are Symmetric Positive Defined (SPD) matrices and naturally reside in a Riemannian manifold, therefore, the traditional Euclidean distance metric is no longer applicable. In other words, in the process of k-Means clustering, when measuring similarity between two features, the traditional Euclidean distance metric will be replaced by the version of Riemannian manifold; that is Stein divergence in our paper. Also, the machine learning algorithms on Euclidean space should be adapted as well. Thus, we encode all RCovDs of one image into a histogram-like feature (vectorized feature) by using Riemannian counterpart of the conventional BoF, taking the geodesic distance of the underlying manifold as the metric.

*l 15: "The histogram feature is extracted by encoding RCovDs of the cloud image blocks with BoF-based codebook" - the term "histogram feature" hasn't been explained yet, what is this? How is it relevant to the technique/results of this paper?*

Reply: A bag of features is a vector of occurrence counts of a vocabulary of local image features, that is, a sparse histogram over the vocabulary. The specific steps of BoF method are as follows (Jégou et al., 2010):

1.  Feature extraction: Extract local image descriptors of each image in the training set.
2.  Quantization: Descriptors are quantized into visual words with the *k*-Means algorithm.
3.  Image representation: An image is then represented by the frequency histogram of visual words obtained by assigning each descriptor of the image to the closest visual word.

[Figure]

Figure 3 Four steps for constructing the bag-of-features for image representation.(Tsai, 2012)

The k-dimensional vector is called histogram feature. BoF describes an image as a vector from a set of local descriptors, and it aggregates the local features to obtain a global histogram representation. In fact, to encode and describe an image with the histogram vector, a good classification result can be achieved. Moreover, BoF has a dimensionality reduction effect, which will be much more effective for the subsequent SVM classification process.

*l 17: "The experiments on the ground-based cloud image datasets validate the proposed method and exhibit the competitive performance against state-of-the-art methods." - this should instead specify exactly what the improvements on previous work is, give the numbers that indicate the improvement and what the implications of these improvements are.*

Reply: Many thanks for the valuable comment. The last sentence of the Abstract will be changed as follows:

"The experiments on the ground-based cloud image datasets shows that a very high prediction accuracy (more than 98% on two datasets) can be obtained with a small number of training samples, which validate the proposed method and exhibit the competitive performance against state-of-the-art methods."

*General:*

*- the section on "Feature extraction" should be before "Region Covariance Descriptors" since the covariance descriptors used the features.*

Reply: Thanks for your reminder. These two sections will be interchanged in the revised manuscript.

*l 102: the relationship between w in the "Rectangular region R with size w x w" and the width of the input image isn't specified.*

Reply: Thanks for the comment. The relationship between *w* and rectangular region *R* is illustrated in **Section 3.1**.

**135** **Once again, thank you very much for your creative comments and suggestions.**

**References**

Chen, T., Rossow, W. B., and Zhang, Y.: Radiative Effects of Cloud-Type Variations, J. Clim., 13, 264-286, 10.1175/1520-0442(2000)013<0264:reoctv>2.0.co;2, 2000.

**140** Hartmann, D. L., Ockert-Bell, M. E., and Michelsen, M. L.: The Effect of Cloud Type on Earth's Energy Balance: Global Analysis, J. Clim., 5, 1281-1304, 10.1175/1520-0442(1992)005<1281:Teocto>2.0.Co;2, 1992.

Jégou, H., Douze, M., and Schmid, C.: Improving Bag-of-Features for Large Scale Image Search, Int. J. Comput. Vision, 87, 316-336, 10.1007/s11263-009-0285-2, 2010.

Ramanathan, V., Cess, R., Harrison, E. F., Minnis, P., Barkstrom, R. B., Ahmad, E., and Hartmann, D.: Cloud-radiative forcing and climate:

**145** Results from the Earth's radiation budget, Science, 243, 57-63, 10.1126/science.243.4887.57, 1989.

Tsai, C.-F.: Bag-of-Words Representation in Image Annotation: A Review, ISRN Artificial Intelligence, 2012, 376804, 10.5402/2012/376804, 2012.